# Tracing Baculovirus AcMNPV Infection Using a Real-Time Method Based on ANCHOR^TM^ DNA Labeling Technology

**DOI:** 10.3390/v12010050

**Published:** 2020-01-02

**Authors:** Aurélie Hinsberger, Benoît Graillot, Christine Blachère Lopez, Sylvie Juliant, Martine Cerutti, Linda A. King, Robert D. Possee, Franck Gallardo, Miguel Lopez Ferber

**Affiliations:** 1LGEI, IMT Mines Alès, Institut Mines-Télécom et Université de Montpellier Sud de France, 6 Avenue de Clavières, 30100 Alès, France; aurelie.hinsberger@gmail.com (A.H.); graillot.benoit@gmail.com (B.G.); christine.blachere-lopez@mines-ales.fr (C.B.L.); 2INRA, SPE, 400 route des Chappes BP 167, 06903 Sophia-Antipolis CEDEX, France; 3CNRS UPS3044 Baculovirus et Thérapie, LabEx-53, 30380 Saint Christol lèz Alès, France; sylvie.juliant@cnrs.fr (S.J.); martine.cerutti@cnrs.fr (M.C.); 4Department of Biological & Medical Sciences, Oxford Brookes University, Gipsy Lane, Oxford OX3 0BP, UK; laking@brookes.ac.uk; 5Oxford Expression Technologies Ltd. BioInnovation Hub, Oxford OX3 0BP, UK; r.possee@oetltd.com; 6NeoVirTech SAS, 1 Place Pierre Potier, 31000 Toulouse, France; 7Institute for Advanced Life Science Technology; ITAV USR3505, 1 Place Pierre Potier, 31000 Toulouse, France

**Keywords:** real-time imaging, fluorescence labelling, baculovirus infection

## Abstract

Many steps in the baculovirus life cycle, from initial ingestion to the subsequent infection of all larval cells, remain largely unknown; primarily because it has hitherto not been possible to follow individual genomes and their lineages. Use of ANCHOR^TM^ technology allows a high intensity fluorescent labelling of DNA. When applied to a virus genome, it is possible to follow individual particles, and the overall course of infection. This technology has been adapted to enable labelling of the baculovirus Autographa californica Multiple NucleoPolyhedroVirus genome, as a first step to its application to other baculoviruses. AcMNPV was modified by inserting the two components of ANCHOR^TM^: a specific DNA-binding protein fused to a fluorescent reporter, and the corresponding DNA recognition sequence. The resulting modified virus was stable, infectious, and replicated correctly in *Spodoptera frugiperda* 9 (Sf9) cells and in vivo. Both budded viruses and occlusion bodies were clearly distinguishable, and infecting cells or larvae allowed the infection process to be monitored in living cells or tissues. The level of fluorescence in the culture medium of infected cells in vitro showed a good correlation with the number of infectious budded viruses. A cassette that can be used in other baculoviruses has been designed. Altogether our results introduce for the first time the generation of autofluorescent baculovirus and their application to follow infection dynamics directly in living cells or tissues.

## 1. Introduction

Baculoviruses are viruses specific to arthropods. Their genome consists of a single circular double-stranded DNA molecule that ranges in size from 80 to 200 kb. In the field, baculovirus transmission is mainly via the oral route. During infection, baculoviruses produce two different virions that contain identical genomes but differ in their structure and function. The budded virus (BV) is involved in systemic infection (cell-to-cell infection) within the host [1]. In the occlusion body (OB), the virions are embedded in a proteinaceous matrix, where the major component is a virus-encoded protein called polyhedrin or granulin, according to the virus considered. OBs are responsible for the transmission between hosts, via the environment. From a taxonomic perspective, the *Baculoviridae* family is presently composed of four genera, *Alphabaculovirus*, *Gammabaculovirus*, and *Deltabaculovirus*, whose OBs contain polyhedrin and embed many virions. In contrast, the *Betabaculovirus* have smaller OBs that contain only one virion within a granulin matrix [2]. During the infection cycle, infected cells usually produce both virus particle types in a sequential way, first the BVs, then the OBs.

Baculoviruses are widely used as biological control agents and as heterologous protein expression systems. The success of baculoviruses as biological control agents relies on the fact that the range of each virus species is usually restricted to a very limited number of hosts, they have no side effect on the environment and are safe for human health [3]. Examples of control of insect pests with baculoviruses are the use of Anticarsia gemmatalis multiple nucleopolyhedrovirus (AgMNPV, *Alphabaculovirus*) in Brazil to control the velvetbean caterpillar on soybean [4] or the Cydia pomonella granulovirus (CpGV, *Betabaculovirus*) to control codling moth on apple and pear trees [5]. Analysis of the factors involved in host range determination requires following each step of the infection process both at the cellular level and at the whole insect level. As the production of OBs is not required for survival in cell culture, in the early 1980s the high level of expression of the polyhedrin gene attracted attention for biotechnology applications [6,7], leading to the generalization of baculoviruses as expression systems [8]. Baculoviruses are common tools in many areas of biotechnology [9], mainly for the production of proteins, both in cell culture or in insect larvae [10]. Modifying the regulation of virus gene expression allows modulating the quantity and the post-translational modifications of the foreign proteins expressed [11].

Many different approaches have contributed to today’s comprehension of the baculovirus lifecycle. Among them, the inclusion of an easy-to-recognize marker in the virus genome has allowed the identification of infected cells when the marker is expressed. Fluorescent markers like green fluorescent protein (GFP) variants have been extensively used. Labelling the virus particles (chemically or by the use of fusion proteins) permit following the entry or the egress of the virus from the cell. However, these two approaches have limitations. Expression in the infected cell requires the activation of the virus cycle, which does not allow following the early steps of infection. Labelling the structural proteins of the virus usually yields a low level signal, and does not allow following the virus cycle between uncoating and virus egress.

The ANCHOR™ technology is based on a bacterial genetic element, the *parABS* system, which allows segregation and plasmid partition [12,13]. The *parABS* system is composed of: *parA* (ATPase activity), *parB* (DNA-binding protein) and *parS* (cis acting DNA sequence). An ANCHOR™ cassette contains both one ANCH target sequence (which comes from *parS*) and the gene encoding the corresponding OR protein (which is the *parB* protein), fused to a fluorescent protein, like GFP; OR-GFP binds to DNA that carries ANCH sequences. Once a stable ANCH/OR-GFP binding is established, a multimerization process occurs, and up to 500 molecules of OR-GFP can bind to the same DNA molecule. Such multimerization results in a signal amplification that allows detection of the target DNA (Figure 1) [14]. ANCHOR™ systems have been used successfully to analyze motion of single genomic loci and DNA double-strand break processing in living yeast [15], or for the assessment of the effect of Ganciclovir on human cytomegalovirus HCMV [16], adenovirus infection and biphasic genome replication [17], and chromatin dynamics during transcription in human cells [14].

In this paper, as a proof of concept, we adapted ANCHOR3™ DNA labeling technology for the tracking of Autographa calinornica Multiple Nucleopolyhedrovirus (AcMNPV) in living cells (Figure 1). First, we created a viral AcMNPV genome containing both the ANCH target sequence and the gene encoding the corresponding OR-GFP protein at two separate loci (AcMNPV1–ANCHOR3). We have verified that the virus obtained is viable, that the fluorescence is high enough to follow a single virus particle, and that it is proportional to the number of particles. Then, we constructed a cassette containing all ANCHOR3™ components in a single plasmid at one genome location. This approach allows for generating recombinant viruses in a single step. In addition, it opens the way for an easy adaptation of this technology to other baculoviruses.

## 2. Materials and Methods

### 2.1. Virus and Cells

A modified AcMNPV was used as a base for the constructs [18]. All final constructs retain the *polh* gene, allowing a positive selection by the presence of occlusion bodies and infectivity for larvae.

*Spodoptera frugiperda* 9 (Sf9) cells (ATCC # CRL1711) [19] were cultured in the laboratory using previously published conditions [19], at 28 °C in TC100 medium supplemented with 5% FBS, 0.5% penicillin-streptomycin and 1% pluronic acid.

### 2.2. Construction of Baculovirus Transfer Vectors

All plasmid constructions were carried out and amplified in *E. coli* XL1 blue bacteria. For the first proof of concept, the two components of the ANCHOR3^TM^ system (provided by NeoVirTech, Toulouse, France) were used. They were obtained from plasmids pANCH3 and OR3GFP, by PCR amplification (Vent^®^ DNA Polymerase, New England Biolabs, Ipswich, MA, USA) of the specific sequences using primers that provide the appropriate cloning restriction sites. The fragment containing the ANCH sequences (982 bps) was amplified and inserted into pGmAc116T [20] between *Eco*RV and *Kpn*I at the *polh* locus. The *or3-gfp* fusion gene-coding sequences (coding the corresponding protein which specifically binds the ANCH3 sequence), fused to *gfp*, (Figure 1) (1786 bps) were amplified using primers providing the appropriate cloning restriction sites. The fragment was inserted into pUC-AD20 (pUC-prA3-stA3) [21] between *Eco*RI and *Mfe*I. To allow continuous expression of the gene upon baculovirus infection of cells, this fragment was inserted downstream of the *Bombyx mori* A3 promoter (prA3) and upstream A3 stop (StA3) (GenBank accession number U49854). The whole cassette containing the A3 promoter, the *or3-gfp* fusion gene, and the A3 terminator was excised with *Xba*I and inserted into the baculovirus transfer vector pVT/gp37, at the *gp37* locus [22]. All constructs were verified by their restriction profiles. Both transfer vector DNAs were purified using NucleoBond^®^ Xtra Midi (Macherey-Nagel, Hoerdt, France).

For the construction of a single plasmid cassette, both components of ANCHOR3^TM^ were inserted in the pAcUW2B transfer vector [23], by substituting the *p10* promoter moiety by the prA3-*or3-gfp*-A3term cassette, then, the ANCH3 sequences were inserted at the *Eco*RV site located upstream of the *polh* promoter. pAcUW2B retains the *polh* gene, allowing a positive selection when co-transfecting with *∆polh* virus DNA.

### 2.3. Construction of Recombinant AcMNPV–ANCHOR3 Baculovirus

AcMNPV1–ANCHOR3 was constructed by inserting the two components of the system via homologous recombination between the parental virus DNA and the two transfer vectors pGmAc116T-ANCH3 and pVT/gp37-OR3. These inserts were placed in the baculovirus genome at the *polh* and at the *gp37* genes (Ac ORF 64), respectively [22] (Figure 2).

Co-transfection was carried out using Sf9 cells and DOTAP (N-[1-(2,3-Dioleoyloxy)propyl]-N,N,N-trimethylammonium methyl-sulfate) as transfection agent. Recombinant viruses, that are fluorescent upon UV light and POLH+ were then selected by plaque assay to obtain pure virus stocks, and then amplified to high titers (2 × 10^7^ plaque forming units PFU/mL) in Sf9 cells. These stocks were stored at 4 °C. The virus titer was determined by plaque assay [24].

AcMNPV2–ANCHOR3 was obtained by co-transfecting pAcUW2B–ANCHOR3 with BacPAK6, a parental virus DNA [25]. Recombinants at the *polh* region were selected by their POLH+ phenotype and by their fluorescence, using the procedure previously described.

Isolation and quantification of viruses were performed by plaque assay. Sf9 cells at 1 × 10^6^ cells·mL^−1^ were plated into 6-well Falcon^®^ plates, in serum-free medium supplemented with 0.5% penicillin-streptomycin and 1% pluronic acid for one hour. The medium was then discarded. Cells were infected with 500 µL of each virus at a range of different concentrations, and incubated during one hour at room temperature (RT); then the inoculum was removed. Two mL of medium with 1% (*w*/*v*) of agarose was added into each well, the plates were then incubated for five days at 28 °C [24].

### 2.4. Spodoptera Exigua Infection

*S. exigua* eggs were provided by P. Caballero, Universidad Pública de Navarra, and reared on artificial diet (Stonefly Heliothis diet. Ward’s Science, Rochester, NY, USA). *S. exigua* larvae (seven days old) were fed with OBs obtained from cell culture by spreading the cellular pellet that contains OBs onto the diet.

### 2.5. Fluorescence Imaging and Quantification

Live microscopy was performed as previously described [16] using a Zeiss Axiovert Observer Z1. Sf9 cells infected with AcMNPV1-ANCHOR3 were observed using a ×63 objective. Image analysis was performed using Fiji software. Visualization of the *S. exigua* larva infected by recombinant AcMNPV was carried out using a ×10 objective.

Fluorescence quantification was performed using 96-well, black flat-bottom plates filled with 100 µL of supernatant of infected cell culture at various times post-infection, using as control supernatant of wildtype (Wt) AcMNPV infected cells. In parallel, each sample titer was determined by plaque assay as indicated previously. The excitation wavelength used was 485 nm and the emission wavelength was 530 nm (bandwidth 20 nm for both). The intensity of fluorescence was measured in triplicate on a fluorescence reader Tecan Spark™ 10M microplate reader using SparkControl V1.2.20 software.

## 3. Results

### 3.1. Construction of AcMNPV–ANCHOR3 Viruses

In a first approach, the two components of the ANCHOR3^TM^ system were inserted at two different loci in the AcMNPV genome, to avoid putative interferences. Co-transfection of the parental AcMNPV bacmid DNA and the transfer vectors containing the two components of the ANCHOR^TM^ system allowed the recovery of recombinant dual viruses that were cloned by plaque purification (Figure 1). The clones obtained were checked by their fluorescence ability, and the insertions were verified by PCR. A second approach was then used to prepare a single cassette allowing the insertion of both components at the same locus (Figure 2). This second virus AcMNPV2–ANCHOR3 behaves similarly to the first; no differences were observed between the two viruses.

### 3.2. Structural and Biological Characterisation of AcMNPV–ANCHOR Virus

The association of a high number (up to 500 copies) of the OR3-GFP fusion protein to the virus genome could interfere with virus nucleocapsid structure. Therefore, the structure of the virions has been analyzed by electron microscopy. No differences could be observed in the overall structure of the virions (Appendix A).

The replication of AcMNPV1-ANCHOR3 virus was observed in vitro and in vivo. Sf9 cells and *S. exigua* larva were infected with the recombinant virus. No delay was observed in the formation of OBs. Observation of the infected cells under fluorescence microscopy confirmed that abundant OBs were produced, and that the fluorescence was concentrated in the OBs, resulting in a characteristic “spotty” appearance (Figure 3a). The fluorescence present in a diffuse form in the cytoplasm is due to the expression of the OR3-GFP protein, that later migrates to the nucleus and condenses with ANCH3 containing virus DNA. At early stages of infection, it is possible to see brilliant spots of fluorescence in the cytoplasm or in the cytoplasmic membrane that correspond to nucleocapsid migration and budded virus egress. The infection of culture cells could be followed under microscope, although the amplitude of the fluorescence variation did not allow a single parameter setting for the whole infection time course (see Appendix A for late times of infection). OBs were used to orally infect third instar (seven days old) *S. exigua* larvae. Forty-eight hours post-infection (hp.i.), the living larvae were observed under a confocal fluorescence microscope. Infection of the tissues was observed, particularly at the level of the tracheal epithelium (Figure 3b,c). Even though the larval cuticle shows autofluorescence with the excitation/emission filters used, the presence of fluorescent particles in the midgut and the tracheal tissues can be observed.

A one-step growth experiment was carried out with wild-type and recombinant viruses (Figure 4). A slightly lower rate of replication and a lower final titer of BV was obtained for recombinant viruses. The final titer for AcMNPV1-ANCHOR3 was 1.24 × 10^7^ OBs/mL compared to 2.07 × 10^7^ OBs/mL for AcMNPV-Wt. Comparable results were obtained for AcMNPV2-ANCHOR3.

Successive passages of virus replication in cell cultures of AcMNPV–Wt and AcMNPV1–ANCHOR3 were carried out (Ai, i corresponding to the passage number). Virus titers at passages A3, A4, and A5 for both viruses were measured by plaque assay (Table 1); mean and standard deviation (SD) are also shown. The titer of AcMNPV–Wt was always slightly higher than that of AcMNPV1–ANCHOR3, in accordance with the results obtained in the growth experiment.

### 3.3. Real-time Visualization of AcMNPV–ANCHOR

The infection of Sf9 cells by AcMNPV1–ANCHOR3 was followed under a fluorescence microscope with a time lapse of 30 min from 0 to 17 hp.i and 10 min from 21 h p.i. to 42 h p.i. The size of a BV particle is at the limit of the resolution of the microscope and thus appears as a single pixel. Specific infection events were captured: nucleocapsids migrating through the cytoplasm, BV budding out of the cell, progeny virus labeled in culture medium, and OBs production in the nucleus (Figure 5).

Representative images of the infection cycle are presented in Figure 6. Due to the high variation in the fluorescence level, it was not possible to obtain the whole virus cycle with a single experiment as the exposure length needed to be reduced at later times of infection. The early infection steps (6 hp.i. to 8 hp.i.) are shown in Figure 6a. New progeny viruses labelled with ANCHOR3™ appeared as fluorescent spots at 6.5 hp.i. in the nucleus, but it was possible to see BV migration in the cytoplasm as single spots. The number of spots in the nucleus increased with time.

The later steps of infection are shown in Figure 6b. At 28 hp.i., multiple OBs were visible, but considerable amounts of uncondensed fluorescence were also present in the nucleus. They correspond to virus genomes that have not been encapsidated. At 29 hp.i., cells became detached, started to expand, and fluorescence looked more diffuse. After 32 hp.i., the cells were dead and free or aggregated OBs were dispersed in the culture medium.

### 3.4. Budded Virus Quantification by Fluorimetry

The relationship between the number of BVs released into the culture medium and the level of fluorescence was analyzed at 0, 8, 16, 24, 36, 48, 72, and 96 hp.i. Relative fluorescence unit (RFU) was plotted versus infection time (Figure 7a). Fluorescence of AcMNPV–Wt was used as a negative control and subtracted from recombinant viruses. In Figure 7a, the RFU from 0 hp.i. to 16 hp.i. follows a lag phase, then enters into an exponential phase until 48 hp.i., which marks the stationary phase.

In order to determine the link between RFU of AcMNPV1–ANCHOR3 and the viral concentration, RFU were plotted versus virus titers obtained by plaque assay (Figure 7b). A correlation of *R*^2^ = 0.98 was obtained. This linearity indicates that the fluorimetric reading can be utilized as a quantitative method. The limit of detection (LoD) of the fluorimetric reading method was around 50 BVs, while 1 BV could be identified by microscopy.

## 4. Discussion

Understanding the process of infection at the cellular level within insects is one of the barriers that prevent further understanding of the progress of the infection and the defense barriers that the insect raises against the virus.

The approaches that are available to the scientific community to address this problem has diversified with time; from the original pathology observations [26] through the modern pathology and histological studies [27], it has been possible to describe the progressive modification of the larval organs. The development of electron microscopy has allowed studying specific steps in the virus/host relationships. Multiple images of virus binding to cells, or nucleocapsid migration from the nucleus through the cell cytoplasm and budding from the cellular membrane, has allowed reconstruction of the temporal process.

Chemical labelling of the virus particles, or detection of these particles with immunolabelling, has permitted capturing views of specific processes, while the use of modified viruses expressing marker genes has allowed identification of infections spreading from one cell to another. One of the limits of all available techniques is, however, the impossibility of following the infection process of a single genome in living cells, from virus entry to the nucleus, to the offspring virus genomes packaging into progeny viruses.

To that aim, labeling the DNA in a specific way becomes necessary. The development of the ANCHOR^TM^ technology, based on the bacterial *parABS* partition system, appeared as an attractive way to attain this goal. The labelling using this approach results in the virus DNA being covered with a relatively high number of OR-GFP proteins. In previous work [14], it has been estimated that the ANCH3 cassette induces the polymerization of up to 500 copies of the OR3-GFP protein. As the baculovirus genome is tightly associated with the P6.9 basic protein and packed into the nucleocapsid mainly constituted of VP39, it was important to assess whether the presence of such proteins would interfere with the virus assembly and thus with the virus infectivity. In addition, the ability of these particles filled with OR3-GFP proteins to occlude into OBs required specific tests. An AcMNPV virus carrying ANCH3 upstream of the *polh* locus and OR3-GFP in lieu of *gfp37* was constructed. Under scanning electron microscopy, no differences were observed between AcMNPV1-ANCHOR3 and the wild-type virus either at the level of the nucleocapsids, or on the global OB structure. The infectivity of the recombinant viruses was tested both for BV in cell culture and for OBs in larva, on a *per os* infection using *S. exigua*. A small decrease in the final titer obtained in cell culture was observed.

The recombinant AcMNPV1–ANCHOR3 virus appears to be stable at least during six successive passages in vitro, suggesting that the bacterial sequences do not suffer from strong selection in the insect system. These results confirm other results showing that a 10^9^ amplification of an HCMV ANCHOR^TM^ virus still leads to the production of 93% of ANCHOR^TM^ tagged viruses in vitro [16].

AcMNPV1–ANCHOR3 was used to follow the cycle of infection in Sf9 cells. The size of the BV, 200 nm × 100 nm, is too small to be resolved by light microscopy. However, the intensity of the fluorescence emission by a single particle was sufficient to be detected as a single pixel by our microscope. It was possible to follow the first contact with the cell, and the internalization of the virus particle.

At early times of infection, the development of the viral stroma can be observed as a diffuse fluorescence, followed soon by more intense zones probably corresponding to the viral factories [28]. These authors indicate that the number of points of DNA replication would be about 15. We observe about this number (Figure 6).

At late times of infection, many bright spots fill the entire nucleus of the infected cell that correspond to the OBs. It should be noted that there is fluorescence outside these OBs, confirming that baculovirus DNA is present in a free form in the nucleus, as previously observed by Vanarsdall et al. [29], which estimated that only about 28% of the viral DNA synthesized was encapsidated into the OBs. The green halo in the cytoplasm is due to the reflection of fluorescence from the nucleus, and not to a high level of OR3-GFP in the cytoplasm. In fact, almost all OR3-GFP is translocated to the nucleus.

The linearity between the fluorescence level and the BV titer suggests that all virus particles are equally decorated by the OR3-GFP protein, and that expression under the *B. mori* A3 promoter was sufficient.

To facilitate future use of the ANCHOR^TM^ system in other baculoviruses, a single cassette containing both components, OR3-GFP and ANCH3, was constructed and a recombinant virus, AcMNPV2–ANCHOR3, was produced. This virus behaves similarly to AcMNPV1–ANCHOR3.

In conclusion, it has been possible to adapt the ANCHOR™ system to the baculovirus. By labelling the virus DNA as soon as it is synthesized, it has been possible to follow a virus particle and its offspring in an infected cell or in a whole larva. This approach opens the way to study the blocking points in resistant insects, or the outcomes in mixed infections.

## Figures and Tables

**Figure 1 viruses-12-00050-f001:**
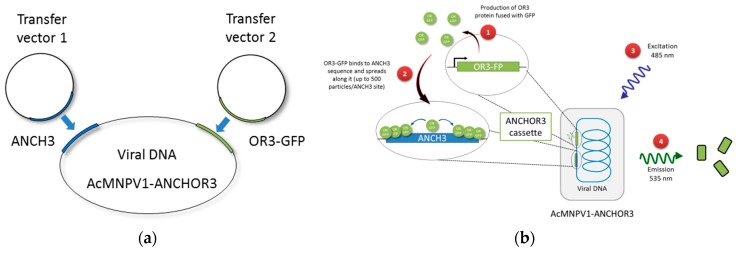
(**a**) Construction and function of the AcMNPV1–ANCHOR3 virus containing both ANCH3 and OR3-GFP. (**b**) Schematic to show function of a ANCHOR3™ virus. (1) OR3-GFP expression; (2) OR3-GFP binding to ANCH3 sequence (up to 500 particles per ANCH3 site); (3) Excitation of viral DNA containing ANCHOR3™ cassette at 485 nm; (4) Light emission at 535 nm.

**Figure 2 viruses-12-00050-f002:**
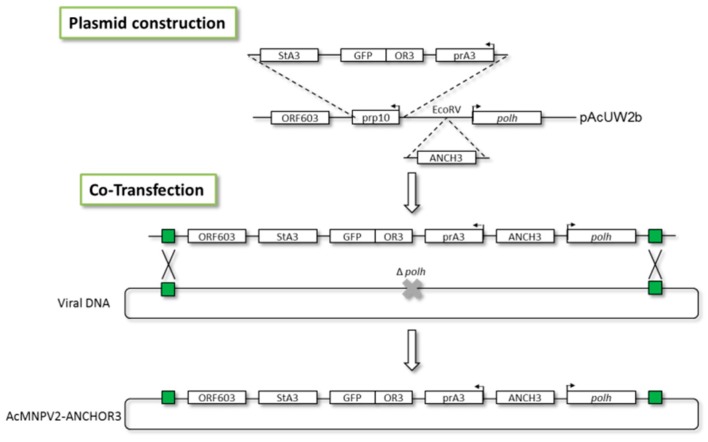
AcMNPV2–ANCHOR3 construction. Both components of the ANCHOR3™ system were inserted into a single transfer vector based on pAcUW2B, where the *p10* promoter region was substituted by the A3-OR3-GFP-STA3. The OR3-GFP sequences were inserted between the *polh* gene and the A3 promoter, at the *Eco*RV site.

**Figure 3 viruses-12-00050-f003:**
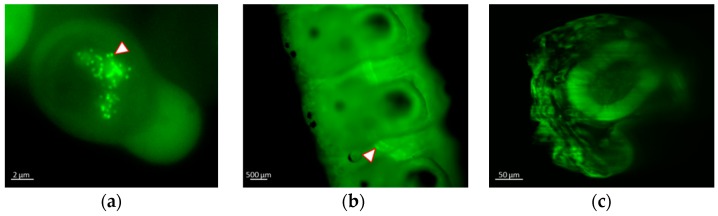
Visualization of recombinant virus infection by fluorescence microscopy using a Zeiss Axiovert Observer Z1 in (**a**) *S. frugiperda* living cells at 18 hp.i. Arrow shows individual OBs (Sf9), (**b**) Alive *S. exigua* larva at 48 hp.i. Arrow shows tracheal epitheium infectedand (**c**) cross-section reconstruction of larva showing midgut infection at 48 h.pi.

**Figure 4 viruses-12-00050-f004:**
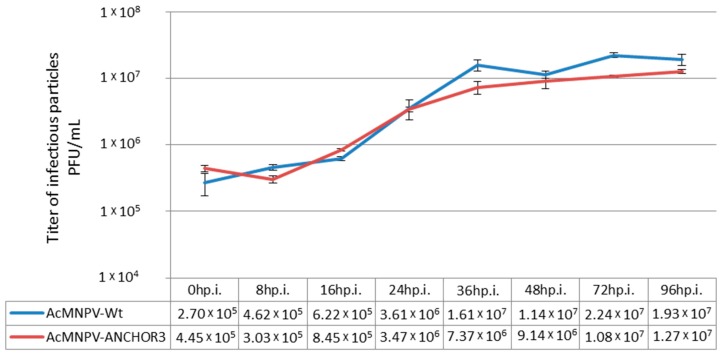
One step growth curve of Wildtype (Wt) and recombinant AcMNPV1–ANCHOR3 virus. The titers correspond to the average of three independent titrations.

**Figure 5 viruses-12-00050-f005:**
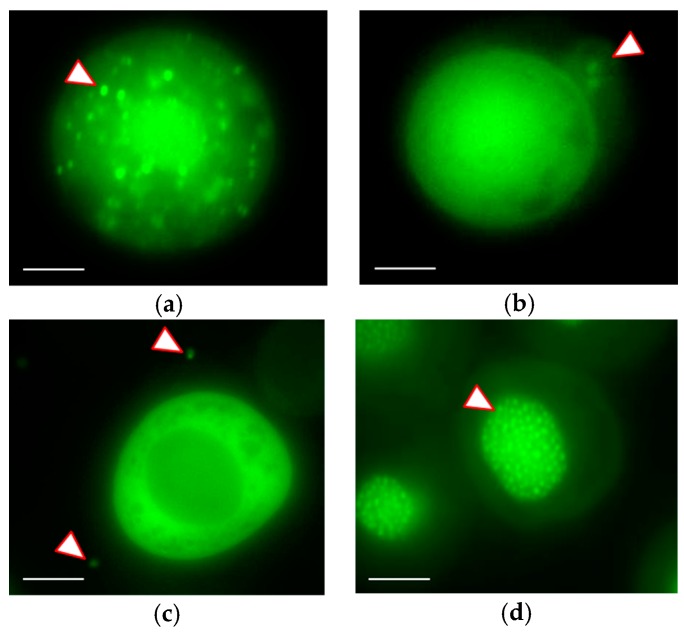
Events of Sf9 cell infection by AcMNPV1–ANCHOR3 visualized with a Zeiss Axiovert Observer Z1 (×63 objective) (**a**) Nucleocapsids (arrow) migrating from the nucleus to the cytoplasmic membrane to release BV; (**b**) BVs (arrow) releasing from infected cell; (**c**) New progeny viruses (arrow) labelled with ANCHOR3^TM^ in the culture medium; (**d**) OBs (arrow) accumulation in the nucleus of the infected cell. Scale bar = 10 µm.

**Figure 6 viruses-12-00050-f006:**
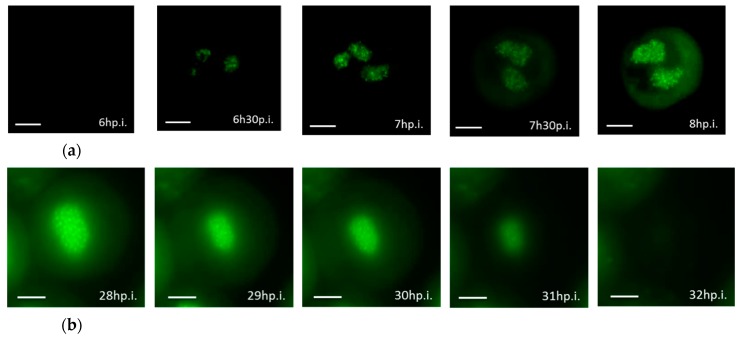
Sf9 cells infection by AcMNPV–ANCHOR visualized with Zeiss Axiovert Observer Z1 (×63 objective) (**a**) early in infection, 6 hp.i. to 8 hp.i; (**b**) late in infection: 28 hp.i. to 32 hp.i. Scale bar = 10 µm.

**Figure 7 viruses-12-00050-f007:**
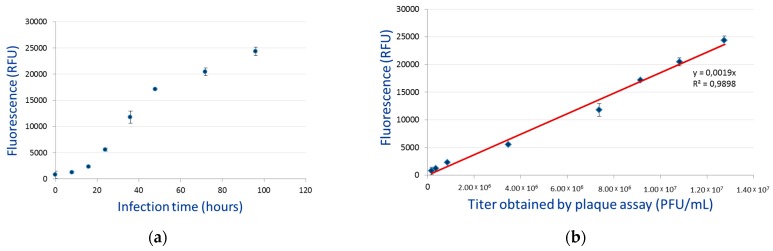
Use of fluorimetry to analyze BV dynamics. (**a**) Fluorimetric quantification of recombinant BV production. (**b**) Correlation between flurometric counts and plaque assay estimation of BV number.

**Table 1 viruses-12-00050-t001:** Titers obtained by plaque assay of different passages (Ai) for AcMNPV–Wt and AcMNPV1–ANCHOR.

Virus	A3	A4	A5	Mean	SD
AcMNPV–Wt	5.80 × 10^7^	9.10 × 10^6^	1.93 × 10^7^	2.88 × 10^7^	2.58 × 10^7^
AcMNPV1–ANCHOR3	2.42 × 10^7^	3.12 × 10^6^	1.27 × 10^7^	1.33 × 10^7^	1.06 × 10^7^

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
