# Peer review of "Tracing Baculovirus AcMNPV Infection Using a Real-Time Method Based on ANCHORTM DNA Labeling Technology"

_viruses, 2020, doi:10.3390/v12010050_

Round 1

Reviewer 1 Report

This manuscript describes a novel real-time monitoring method based on ANCHOR DNA labeling technology. The data provided are solid and I feel it can be published without major revisions.

line 180, please include TEM data in the manuscript. Fig. 3: Background GFP level seems very high in infected larvae. Please discuss this reason and add the opinion whether this reporter is sufficient for virus monitoring within the larvae. line273, ref#24 is correct? The full sequence of the reporter cassette in Fig. 2 should be given for the readers as a supplement.

Author Response

Dear reviewer,

We appreciate your critical review to our manuscript. We have taken into account your suggestions that contributed to improve our manuscript.

Please find below à point by point answer.

This manuscript describes a novel real-time monitoring method based on ANCHOR DNA labeling technology. The data provided are solid and I feel it can be published without major revisions.

Point 1: line 180, please include TEM data in the manuscript.

Response 1: We accept the comment and add the following figure in the supplementary material

Supplementary Figure 1: Electron microscopy observations of AcMNPV1-ANCHOR3 (a) Scanning electron microscopy of OBs and (b) Transmission electron microscopy of ODVs

Point 2: Fig. 3: Background GFP level seems very high in infected larvae. Please discuss this reason and add the opinion whether this reporter is sufficient for virus monitoring within the larvae.

Response 2: We agree with reviewer. There are two different situations. In Figure 3a, the bhackground GFP level present in the cytoplasm results from the expression of OR3-GFP before its internalization into the nucleus.In Figure b and c, the background is due to the combination of two phenomena: the budded viruses present in the hemolymph, that resemble to a background at low resolution; and a non-specific fluorescence of the chitin that is not intercepted by our filter combination. This point is addressed in the technical article http://zeiss-campus.magnet.fsu.edu/articles/spectralimaging/considerations.html. Although it is possible to subtract this background autofluorescence, we preferred not to apply additional conversions to the pictures.

Point 3: line273, ref#24 is correct?

Response 3: The reviewer is right; there was an error in the references. The reference 24 was changed for:

Germier, T.; Kocanova, S.; Walther, N.; Bancaud, A.;, Shaban, H.A.; Sellou, H.; Politi, A.Z.; Ellenberg, J.; Gallardo, F.; Bystricky, K. Real-time chromatin dynamics at the single gene level during transcription activation. bioRxiv 2017, 111–179

Point 4: The full sequence of the reporter cassette in Fig. 2 should be given for the readers as a supplement.

Response 4: We consider that including the precise sequences of the construction does not present an added value to our work. The appropriate information can be found in patent number (FR3081474) and has been previously described in the literature cited.

Reviewer 2 Report

This manuscript (Viruses 673599) describes the application of a relatively new DNA labelling technology previously developed in other systems for the purposes of real time tracking of baculovirus infections in cultured cells and host insect larvae.

Significant Issues:

Section 3.1 – Reference to Figure 2 cites Fig 2a and Fig 2b but on the actual figure there is no designated panel (a) and the Figure legend does not mention either panel (a) or (b). This makes the results in lines 168-173 harder to follow than it should be.

Section 3.2 – Lines 185-190 – Results from 3rd instar infections are described. It is very difficult from Fig (b) to distinguish infection in particular cell types and specifically at the “level of trachea (and should this not be “tracheal epithelium”). There is no information in the M&M or the Results section as to the methods or approach used to generate “three dimensional reconstruction of larvae” from serial observations. Finally lines 189-190 refers to Supplemental Video S1 to virus tagged fluorescence in midgut and trachea; however, this video shows time lapse in Sf9 infected cells so I fail to see how this can be used to support this statement on location of infected tissue in larvae.

Section 3.3 – Lines 215-216 – Figure 5 is used to demonstrate infection events including “,OBs production and release (Figure 5).” While Fig. 5 (c) may show OB production in infected nuclei, Fig. (d) show BV egress from cells not OB release. A magnification/size bar on Figures 5 and 6 as shown on Figure 3 would also be helpful.

Section 3.4 – Lines239-240 – A more complete description of the derivation of RFU either here or in the M&M section would be helpful.

Minor Points:

Line 43 – “responsible of the transmission” should read “responsible for the transmission”

Line 50 – “facts” should read “fact”

Author Response

Dear reviewer,

We appreciate your critical review to our manuscript. We have taken into account your suggestions that contributed to improve our manuscript.

Please find below à point by point answer.

This manuscript (Viruses 673599) describes the application of a relatively new DNA labelling technology previously developed in other systems for the purposes of real time tracking of baculovirus infections in cultured cells and host insect larvae.

Point 1: Section 3.1 – Reference to Figure 2 cites Fig 2a and Fig 2b but on the actual figure there is no designated panel (a) and the Figure legend does not mention either panel (a) or (b). This makes the results in lines 168-173 harder to follow than it should be.

Response 1: Thanks for the correction. We accept the modification. The paragraph now reads as follows:

In a first approach, the two components of the ANCHOR3TM system were inserted at two different loci in the AcMNPV genome, to avoid putative interferences. Cotransfection of the parental AcMNPV bacmid DNA and the transfer vectors containing the two components of the ANCHORTM system allowed the recovery of recombinant dual viruses that were cloned by plaque purification (Figure 1). The clones obtained were checked by their fluorescence ability, and the insertions were verified by PCR. A second approach was then used to prepare a single cassette allowing the insertion of both components at the same locus (Figure 2). This second virus AcMNPV2-ANCHOR3 behaves similarly to the first, no differences were observed between the two viruses.

Point 2: Section 3.2 – Lines 185-190 – Results from 3rd instar infections are described. It is very difficult from Fig (b) to distinguish infection in particular cell types and specifically at the “level of trachea (and should this not be “tracheal epithelium”). There is no information in the M&M or the Results section as to the methods or approach used to generate “three dimensional reconstruction of larvae” from serial observations. Finally lines 189-190 refers to Supplemental Video S1 to virus tagged fluorescence in midgut and trachea; however, this video shows time lapse in Sf9 infected cells so I fail to see how this can be used to support this statement on location of infected tissue in larvae.

Response 2: We have modified the « infection of the trachea » to infection of the tracheal epithelium » as requested. We have deleted the reference to the three dimensional reconstruction of the larvae, as the video was lost. We added more information, requested by reviewer 1, on the background luorerscence. The paragraph now reads as follows:

“The replication of AcMNPV1-ANCHOR3 virus was observed in vitro and in vivo. Sf9 cells and S. exigua larva were infected with the recombinant virus. No delay was observed in the formation of OBs. Observation of the infected cells under fluorescence microscopy confirmed that abundant OBs were produced, and that the fluorescence was concentrated in the OBs, resulting in a characteristic 'spotty' appearance (Figure 3a). The fluorescence present in a diffuse form in the cytoplasm is due to the expression of the OR3-GFP protein, that later migrates to the nucleus and condense with ANCH3 containing virus DNA. At early stages of infection, it is possible to see brilliant spots of fluorescence in the cytoplasm or in the cytoplasmic membrane that correspond to nucleocapsid migration and budded virus egress. The infection of cultures cells could be followed under microscope, although the amplitude of the fluorescence variation did not allow a single parameter setting for the whole infection time course (see movie S1 in supplementary material for late times of infection). OBs were used to orally infect 3rd instar (7 days old) S. exigua larvae. Forty-eight hours post-infection (hp.i.), the living larvae were observed under a confocal fluorescence microscope. Infection of the tissues was observed, particularly at the level of the tracheal epithelium (Figure 3b, 3c). Even though the larval cuticle shows autofluorescence with the excitation/emission filters used, the presence of fluorescent particles in the midgut and the trachea can be observed.”

Point 3: Section 3.3 – Lines 215-216 – Figure 5 is used to demonstrate infection events including “,OBs production and release (Figure 5).” While Fig. 5 (c) may show OB production in infected nuclei, Fig. (d) show BV egress from cells not OB release. A magnification/size bar on Figures 5 and 6 as shown on Figure 3 would also be helpful.

Response 3: Scale bar was added for Figure 5 and 6. We fully agree with the reviewer. Figure 5 does not refer to OB production and release. In this experiment, all cells were not infected at the same time, and so, the h p.i. indicated induces an error. Accordingly we have deleted the time and inverted the panels to follow a logical progression.

Point 4: Section 3.4 – Lines239-240 – A more complete description of the derivation of RFU either here or in the M&M section would be helpful.

Response 4: in M&M we precise our method. The paragraph now reads:

Fluorescence quantification was performed using 96-well, black flat-bottom plates filled with 100µL of supernatant of infected cell culture. In parallel, each sample titer was determined by plaque assay as indicated previously, .

Point 5: Line 43 – “responsible of the transmission” should read “responsible for the transmission”

Response 5: The sentence has been modified as requested

Point 6: Line 50 – “facts” should read “fact”

Response 6: The sentence has been modified as requested

Reviewer 3 Report

In this report the application a new technique to trace viral genomes is described for a baculovirus.  They suggest that it may be useful in the investigation of DNA production in viral mutants.  It could also be used for the study of virus DNA replication check points in non-permissive cells.  The manuscript could be improved if the suggestions below are incorporated into a revised manuscript.

36 …

 Their genome consists of a single circular double-stranded DNA…

39 …

 a burgeoning form of the virus

Fig 1b.  This appears to show the virus with an icosahedral morphology which is misleading since the virions are rod-shaped. 

190  I could not find the S1 movie of an infected larva in the supplemental materials.  There is a single movie showing what appear to be infected cells.

193  …(b) S. exigua larva …  Is this a whole larva or a specific tissue?

The times pi for each panel should be indicated.

Early in the infection of the midgut of larvae, were they able to observe virions that appeared to transit the cells without replication as has been suggested?

199 isn’t 2 x 107 somewhat low for a wt AcMNPV?

Fig 7B.  It would be easier to interpret if the titers were shown in powers of 10.  Counting all those zeros is very challenging.

288 Reference 17 is to adenovirus not HCMV

Author Response

Dear reviewer,

We appreciate your critical review to our manuscript. We have taken into account your suggestions that contributed to improve our manuscript.

Please find below à point by point answer.

In this report the application a new technique to trace viral genomes is described for a baculovirus.  They suggest that it may be useful in the investigation of DNA production in viral mutants.  It could also be used for the study of virus DNA replication check points in non-permissive cells.  The manuscript could be improved if the suggestions below are incorporated into a revised manuscript.

Point 1: 36 …Their genome consists of a single circular double-stranded DNA…

Response 1: The sentence has been modified as requested

Point 2: 39 … a burgeoning form of the virus

Response 2: We have deleted these words that repeated the defection. The sentence now reads:

The budded virus (BV) is involved in systemic infection (cell-to-cell infection) within the host.

Point 3: Fig 1b.  This appears to show the virus with an icosahedral morphology which is misleading since the virions are rod-shaped. 

Response 3: Figure 1b was modified. We hope that now it is cleared.

Point 4: 190  I could not find the S1 movie of an infected larva in the supplemental materials.  There is a single movie showing what appear to be infected cells.

Response 4: The reviewer is right, and the same question was also addressed by reviewer 1. Finally,; we will not present the three dimensional reconstruction. We have modified the text. Now it reads as follows:

“The replication of AcMNPV1-ANCHOR3 virus was observed in vitro and in vivo. Sf9 cells and S. exigua larva were infected with the recombinant virus. No delay was observed in the formation of OBs. Observation of the infected cells under fluorescence microscopy confirmed that abundant OBs were produced, and that the fluorescence was concentrated in the OBs, resulting in a characteristic 'spotty' appearance (Figure 3a). The fluorescence present in a diffuse form in the cytoplasm is due to the expression of the OR3-GFP protein, that later migrates to the nucleus and condense with ANCH3 containing virus DNA. At early stages of infection, it is possible to see brilliant spots of fluorescence in the cytoplasm or in the cytoplasmic membrane that correspond to nucleocapsid migration and budded virus egress. The infection of cultures cells could be followed under microscope, although the amplitude of the fluorescence variation did not allow a single parameter setting for the whole infection time course (see movie S1 in supplementary material for late times of infection). OBs were used to orally infect 3rd instar (7 days old) S. exigua larvae. Forty-eight hours post-infection (hp.i.), the living larvae were observed under a confocal fluorescence microscope. Infection of the tissues was observed, particularly at the level of the tracheal epithelium (Figure 3b, 3c). Even though the larval cuticle shows autofluorescence with the excitation/emission filters used, the presence of fluorescent particles in the midgut and the tracheal tissues can be observed.”

Point 5: 193  …(b) S. exigua larva …  Is this a whole larva or a specific tissue?

The times pi for each panel should be indicated.

Response 5: Time after infection has been indicated for each panel in Figure 3. We have also indicated that both cells and larvae were alive. Accordingly, it is the whole larva that was observed on the fluorescence microscope.

Point 6: Early in the infection of the midgut of larvae, were they able to observe virions that appeared to transit the cells without replication as has been suggested?

Response 6: As we are using the whole living larvae, we have not yet managed to reach the good resolution. This would be possible using dissected midguts. This is one of the next goals.

Point 7: 199 … isn’t 2 x 107 somewhat low for a wt AcMNPV?

Response 7: When we performed the infection, we obtained this titer, we were not trying to reach the optimal conditions of a viral production.

Point 8: Fig 7B.  It would be easier to interpret if the titers were shown in powers of 10.  Counting all those zeros is very challenging.

Response 8: The Figure has been modified as requested.

Point 9: 288 Reference 17 is to adenovirus not HCMV

Response 9: We accept the modification and change the reference for:

Mariamé, B.; Kappler-Gratias, S.; Kappler, M.; Balor, S.; Gallardo, F.; Bystricky, K.; Jung, J.U. Real-time visualization and quantification of human cytomegalovirus replication in living cells using the ANCHOR DNA labeling technology. J. Virol. 2018, 92, 571–589